# A Recreational Swimming Intervention during the Whole School Year Improves Fitness and Cardiometabolic Risk in Children and Adolescents with Overweight and Obesity

**DOI:** 10.3390/ijerph192417093

**Published:** 2022-12-19

**Authors:** Elisabeth Machado, Fernanda Jannuzzi, Silvio Telles, Cecilia Oliveira, Isabel Madeira, Fernando Sicuro, Maria das Graças Souza, Alexandra Monteiro, Eliete Bouskela, Paulo Collett-Solberg, Paulo Farinatti

**Affiliations:** 1Clinical and Experimental Research Laboratory on Vascular Biology, Biomedical Center, University of Rio de Janeiro State, Rio de Janeiro 20550-900, Brazil; 2Department of Pediatrics, Faculty of Medical Sciences, University of Rio de Janeiro State, Rio de Janeiro 20550-900, Brazil; 3Laboratory of Physical Activity and Health Promotion, Institute of Physical Education and Sports, University of Rio de Janeiro State, Rio de Janeiro 20550-900, Brazil; 4Institute of Nutrition, University of Rio de Janeiro State, Rio de Janeiro 20550-900, Brazil; 5Centre for Environment and Marine Studies, Department of Biology, University of Aveiro, 3810-193 Aveiro, Portugal; 6Department of Radiology, Faculty of Medical Sciences, University of Rio de Janeiro State, Rio de Janeiro 20550-900, Brazil

**Keywords:** physical activity, cardiovascular risk, blood pressure, endothelial function, autonomic modulation, health

## Abstract

The benefits of swimming as a treatment for overweight children are undefined. We investigated the effects of recreational swimming on cardiometabolic risk in children/adolescents with normal and excess weight. Participants (*n* = 49, 26 girls, 10.3 ± 1.8 y) were grouped as ‘eutrophic swimming’ (EU-Swim, *n* = 14); ‘excess weight swimming’ (EW-Swim, *n* = 20) with an ‘obese swimming’ subgroup (OB-Swim, *n* = 10); and ‘excess weight sedentary’ (EW-Sed, *n* = 15) with an ‘obese sedentary’ subgroup (OB-Sed, *n* = 11). Swimming (50 min, twice/week, moderate-vigorous intensity) was an extra activity during the school year (6 + 3 months with a 3-month school break). Nutritional status, blood pressure (BP), physical activity, cardiorespiratory fitness, biochemical variables, autonomic modulation, endothelial function, abdominal fat, and carotid thickness were assessed at baseline, 6, and 12 months. Greater improvements (*p* < 0.05) occurred in EW-Swim vs. EW-Sed in body mass index (z-BMI, −16%, *d+* 0.52), waist-to-height ratio (W/H, −8%, *d+* 0.59–0.79), physical activity (37–53%, *d+* 1.8–2.2), cardiorespiratory fitness (30–40%, *d+* 0.94–1.41), systolic BP (SBP, −6–8%, *d+* 0.88–1.17), diastolic BP (DBP, −9–10%, *d+* 0.70–0.85), leptin (−14–18%, *d+* 0.29–0.41), forearm blood flow (FBF, 26–41%, *d+* 0.53–0.64), subcutaneous fat (SAT, −6%, *d+* 0.18), and intra-abdominal fat (VAT, −16%, *d+* 0.63). OB-Swim showed improvements vs. OB-Sed in TNFα (−17%, *d+* 1.15) and adiponectin (22%, *d+* 0.40). Swimming improved fitness and cardiometabolic risk in children/adolescents with overweight/obesity. (TCTR20220216001)

## 1. Introduction

Low physical activity levels have been reported in pediatric populations, which is acknowledged as a major health risk factor [1]. The combination of physical inactivity and excess weight in school-age children increases their cardiovascular risk [2] and the odds of sedentary behavior and obesity in adult life [3,4]. On the other hand, regular physical exercise promotes health benefits decreasing the chances of cardiovascular events in adulthood [5].

Engagement in physical activity has many determinants and involves socioeconomic, cultural, and environmental factors, in addition to genetic, physiological, and psychological aspects [6]. Adherence to regular exercise can be difficult, particularly among children and adolescents with excess weight [7]. Issues related to poor motor proficiency, stigmatization, and stereotypes may affect their motivation and adherence [8]. This is a primary concern when planning physical activity interventions. Traditional and individualized exercise approaches may be monotonous, consisting of excessively systematized and analytical modalities [9]. Sports and recreational activities might be more motivational, with greater potential for long-term adherence. However, current research has neglected how these types of exercise interventions are capable of improving markers of fitness and overall cardiometabolic risk in children and adolescents with excess weight [9].

Previous trials showed that different types of sports activities (e.g., soccer or judo) are capable of improving health markers in school-age children [10,11]. Aquatic exercises are frequently recommended for this age group, given their good acceptance and motivation characteristics. Swimming is traditionally indicated as a general exercise with the potential to develop the strength of several muscle groups while eliciting intensities compatible with cardiorespiratory gains [12]. Moreover, the aquatic environment provides an inviting environment for individuals with excess weight. The water’s buoyancy can help maintain stability compared with ground exercises. Exercising in water provides calorie loss with less fatigue and comfort due to the reduction in body weight, which explains why aquatic modalities have been recommended as an alternate exercise to land exercise for individuals with excess weight [13]. Although children and adolescents adhere well to swimming practice [14], we located a single report showing that a relatively short 12-week intervention improved the physical fitness and vascular compliance of obese elementary students [15]. Additional information is, therefore, necessary for a better understanding of the potential contribution of longer recreational swimming programs to improve the overall health of pediatric populations. Research investigating the impact of more prolonged swimming interventions as typically performed by children and adolescents in actual life would increase the ecological validity of the obtained data. This information might be useful for practitioners and medical doctors that usually indicate this sport modality for children and adolescents.

Given this gap in the literature, the present trial investigated the effects of a recreational swimming intervention applied as an extra activity during the whole school year (6 + 3 months interspersed with a 3-month school break) on the physical activity level, cardiorespiratory fitness, and multiple cardiometabolic health markers (body mass and composition, blood pressure, endothelial function, autonomic control, glucose and insulin, cytokines, intra- and subcutaneous abdominal fat, and carotid intima–media thickness) in children and adolescents with normal and excess weight (e.g., overweight or obesity). We hypothesized that improvements in fitness and cardiovascular risk would occur regardless of changes in body mass or composition among participants exhibiting excess vs. normal weight undergoing swimming intervention vs. physically inactive controls.

## 2. Materials and Methods

### 2.1. Subjects

This was a 12-month controlled trial including 49 school-aged children (26 girls) aged 7 to 16 years (14 with normal weight, 14 with overweight, and 21 with obesity). Nutritional status was defined according to World Health Organization (WHO) standardized BMI (z-BMI) values for age and sex (>+2 z-scores for obesity; >+ 1 and ≤+ 2 for overweight; ≥−2 and ≤+1 for normal weight) [16], calculated using the Anthro Plus^TM^ software version 1.04 (WHO, Geneva, Switzerland). Exclusion criteria were any contraindication to exercise; health problems impairing microcirculatory assessment; kidney, hematological, hepatic, rheumatologic, cardiovascular, respiratory, or endocrine disease; and use of anti-inflammatory medication.

The participants were initially recruited at the pediatric outpatient clinic of a public hospital and through leaflets advertising the swimming project distributed in the University facilities. Figure 1 summarizes the procedures for the sample selection, depicting the number of participants after stratification according to intervention and z-BMI: ‘eutrophic swimming group’ (EU-Swim, *n* = 14), including subjects with normal weight, and ‘excess weight swimming group’ (EW-Swim, *n* = 20), including those with overweight or obesity. This last group underwent further analysis including only participants with obesity (‘obese swimming group’, OB-Swim, *n* = 10). The non-exercise group included children with excess weight who did not engage in any systematic physical activity during the experiment, named ‘excess weight sedentary’ (EW-Sed, *n* = 15). A subgroup including obese individuals was also selected, referred to as ‘obese sedentary’ (OB-Sed, *n* = 11).

Children and adolescents were divided into two groups according to their initial level of aquatic development (Figure 2). Group 1 (G1, *n* = 15): (a) aquatic adaptation stage, focusing on the ambiance, adaptation, breathing, and ability to float and perform movements autonomously, with a duration of 8 weeks and an average covered distance of 200–300 m per class; (b) aquatic stimulation stage, aimed at developing water motor skills, lasting 12 weeks with average covered distance of 300–400 m per class; (c) aquatic domain stage, characterized by mastering the fundamentals of swimming styles, lasting 4 weeks plus 12 weeks after school break with average covered distance of 600–700 m per class. Group 2 (G2, *n* = 19) started in the stimulation phase for 4 weeks and covered an average distance of 350 m per class. Subsequently, the domain stage lasted 16 weeks with 500–800 m per class. Finally, the mastery stage aimed at improving swimming techniques, lasting 4 weeks plus 12 weeks after school break with covered distances of 800–1200 m per class.

The experimental procedures were conducted according to ethical guidelines of the Declaration of Helsinki. All children and their legal guardians provided written informed consent. The study received approval from the local Research Ethics Committee. The trial was registered at a World Health Organization credited office (TCTR20220216001) and conforms to the Consolidated Standards of Reporting Trials (CONSORT) guidelines.

### 2.2. Experimental Design

Data collection occurred during regular school days: Visit 1—clinical and anthropometric assessments; Visit 2—blood collection; Visit 3—resting blood pressure, endothelial function, and heart rate variability (HRV); Visit 4—abdominal and carotid ultrasound; Visit 5—physical activity questionnaire and Yoyo test. The swimming intervention began within 15 days after baseline assessments and consisted of 50 min sessions performed twice a week for 12 months, including aquatic adaptation, fluctuation, breath, and displacement exercises, in addition to techniques of the four swimming style exercises. An initial assessment verified the level of swimming development of participants to organize groups matched for their skills and ensure safety. Swimming classes were performed with moderate-to-vigorous intensity, in a progression that increased in direct proportion to the mastery of the techniques of the swimming styles (please refer to Figure 2). The covered distances in each class were recorded to provide insight into the exercise volume applied in each group and the progression phase of the swimming program.

Two licensed instructors supervised the swimming sessions. Intermediate assessments were performed after 6 months (or 24 weeks) of intervention, except for ultrasound assessments. After this first follow-up, the training had to be discontinued because of the summer school break (3 months or 12 weeks), and then participants resumed the swimming intervention for another 12 weeks (3 months). At the end of the experiment, assessments performed at baseline were repeated. The same trained researchers applied all tests.

### 2.3. Clinical and Anthropometric Outcomes

Body mass was measured to the nearest 0.1 Kg using a digital scale (Tech Line^TM^, São Paulo, SP, Brazil), and height was determined to the nearest 0.1 cm using a tape attached to the wall and a set square. BMI (kg/m^2^) and z-BMI were calculated [16]. Waist circumference was assessed with a flexible tape (nearest 0.1 cm), at the midpoint between the lowest rib and iliac crest, and the waist-to-height ratio (W/H) was calculated. The pubertal maturation stage was assessed according to criteria proposed by Tanner [17], and the presence of *acanthosis nigricans*, which is a dermatological manifestation often concomitant with the presence of hyperinsulinemia was evaluated according to the Burke scale [18].

### 2.4. Physical Activity Level and Cardiorespiratory Fitness

The physical activity level was estimated with the Physical Activity Questionnaire for Older Children (PAQ-C), quantifying moderate-to-vigorous physical activities practiced at school and during leisure time in the previous seven days [19,20]. Answers were rated with a 5-point scale (1 = did not practice; 5 = practiced every day), and the final score corresponded to the average of all responses (1—very sedentary; 2—sedentary; 3—moderately active; 4—active; 5—very active). Screen time was also assessed by the PAQ-C, which includes a question about the average daily hours dedicated to any type of screen: TV, computer, video games, and especially smartphones. It is self-reported and participants responded with the help of their parents. The Yoyo endurance test ‘level 1’ was applied to estimate cardiorespiratory fitness [10,21]. This field test consists of 20 m shuttles with progressively increasing pace interspersed with 5 s active recovery. The aim was to perform as many shuttles as possible and the total distance covered until fatigue was recorded.

### 2.5. Biochemical Analysis

Venous blood samples were collected in the morning after 12 h fasting, being immediately centrifuged and frozen at −80 °C for posterior analysis. Insulin was measured in gamma-C12 equipment with Coat-A-Count radioimmunoassay solid phase (DPC^TM^, Los Angeles, CA, USA). Serum glucose, total cholesterol (TC), high-density lipoprotein (HDL), low-density lipoprotein (LDL), and triglycerides (TG) were quantified through automated enzymatic methods. Leptin, adiponectin, and tumor necrosis factor-alpha (TNF-α) were measured using ELISA kits (R & D Systems^TM^, Minneapolis, MN, USA). Normative references were fasting insulin ≤ 15 μU/mL for prepubescent and ≤20 μU/mL for pubertal individuals; glucose < 100 mg/dL; TG and LDL < 100 mg/dL; and HDL ≥ 45 mg/dL [22].

### 2.6. Blood Pressure, Peripheral Arterial Tonometry, Venous Occlusion Plethysmography, and Heart Rate Variability

The blood pressure was measured using an Omron 705IT oscillometer device (Onrom^TM^, Matsusaka, Japan) after 5 min of rest. The average of three successive measurements interspersed with 3 min was recorded as result. Participants were classified as showing normal blood pressure when systolic and diastolic values were lower than the 90th percentile for sex, age, and height [23].

The endothelial function was non-invasively evaluated by assessing the vasodilatation response through peripheral arterial tonometry (EndoPAT^TM^ 2000, Itamar Medical Ltd., Caesarea, Israel) [24]. Testing was performed after 5 min at rest in a controlled temperature room (22 °C to 24 °C) with participants in a supine position. The forearm blood flow (FBF) was assessed in the left arm through venous occlusion plethysmography (EC-6; Hokanson^TM^, Bellevue, WA, USA) in a temperature-controlled (20–22 °C) room, with subjects at the supine position after 4 h fasting [25]. A mercury-filled strain gauge was placed on the maximal diameter of the upper third forearm, with two inflatable cuffs on the arm and wrist. The average of four cycles was adopted to determine resting FBF (ml/min/100 mL tissue).

In addition, endothelial-dependent vasodilatation was assessed through FBF during post-occlusive reactive hyperemia. FBF was assessed after 3 min arterial occlusion, and values normalized per unit of blood pressure were used to estimate forearm vascular conductance. The percentage increase in flow per minute (% Increment) and the natural logarithmic transformed reactive hyperemia index (LnRHI) were calculated as markers of endothelial function. Values of LnRHI <0.51 were considered indicative of peripheral endothelial dysfunction, as proposed elsewhere [24,26]. The EndoPAT^TM^ device also measured the HRV, and selected indices were used to characterize the parasympathetic modulation at rest: average of coupling intervals of all consecutive normal beats (RR) and the square root of the mean of squares of the differences between successive normal RR intervals (RMSSD) [27].

### 2.7. Ultrasound Assessments

The carotid intima–media thickness (cIMT) and abdominal fat distribution were measured through ultrasound (Aplio XG SSA-790, Toshiba^TM^, Tochigi-ken, Japan) by a single trained professional and with participants in a supine position. A 3.5 MHz convex transducer was used to measure the thickness of intra-abdominal fat (VAT), and a linear multi-frequency transducer was used to measure the thickness of the subcutaneous abdominal wall (SAT). The transducer was positioned perpendicularly without exerting pressure on the abdomen, in a region 1 cm superior to the umbilical scar (xiphoumbilical middle line). SAT and VAT were directly determined through frozen images in the axial plane. SAT was defined as the distance between the skin and the anterior face of the alba line and VAT as the distance between the posterior face of the alba line and the anterior wall of the abdominal aorta [28]. The average of three consecutive measurements was used.

The cIMT was defined as the distance between the luminal side of the endothelium and the distal muscular interface layer of the carotid artery measured at the common carotid arteries, 1.5–2.0 cm below the carotid bifurcation. The average of three consecutive measurements determined the thickness of each side. Values ≥0.45 mm for children under ten years old or ≥0.55 mm for children between ten and 18 years old were considered above normal. The coefficients of variation of those measurements were lower than 3%, which agrees with prior research [29].

### 2.8. Statistical Analyses

Data normality was tested considering the whole data series for each outcome within a given group, using the Shapiro–Wilk test complemented by standard normal probability plots. The W-value rejected the normality assumption for a few variables (triglycerides, leptin, adiponectin, RMSSD, %Increment, and cIMT) but never in all groups. However, in those cases, the plot of z-values computed from rank-ordered deviations from the mean (residuals) *vs*. expected Gaussian values produced a general lack of fit with data forming a clear S shape pattern. The assumption of normal distribution was therefore considered sufficient to perform parametric tests.

The results are presented as mean ± standard deviation. Differences between groups at baseline were tested by 1-way ANOVAs. Pre vs. post comparisons within–between groups were made by 2-way ANOVAs with repeated measures (6 and 12 months). In all cases, the ANOVAs were followed by Fisher post hoc tests in the event of significant *F* ratios. Cohen’s effect sizes (*d+*) were calculated for the significant differences within groups, being classified as small (*d+* = 0.2), medium (*d+* = 0.5), and large (*d+* = 0.8). All calculations were performed using the Statistica 10.0 software (Statsoft^TM^, Tulsa, OK, USA), and the significance level was set at *p* ≤ 0.05.

## 3. Results

Table 1 presents the sample characteristics. Participants had similar ages and were mostly pre-pubertal or at early puberty. Blood pressure was classified as elevated in approximately 30% of participants in EU-Swim and EW-Swim and 55% of those in EW-Sed (one overweight and seven obese). *Acanthosis nigricans* was present in 25% of children with excess weight that enrolled in the swimming intervention and 33% of those in the physically inactive group.

Table 2 presents data on anthropometric outcomes, physical activity level, screen time, cardiorespiratory fitness, and cardiometabolic markers at baseline and after 6 and 12 months in swimming groups vs. physically inactive controls. As expected, at baseline, the lowest z-BMI and W/H were found in EU-Swim and the highest in EW-Sed. Improvements in z-BMI occurred in EU-Swim (62%, *d+* 0.22) and EW-Swim (−16%, *d+* 0.52), while W/H slightly decreased only in EW-Swim (−8%, *d+* 0.59–0.79).

In regards to cardiometabolic risk outcomes, consistent decreases in SBP and DBP occurred in EU-Swim (−9%, *d+* 0.93–1.01, and −10%, *d+* 0.86, respectively) and EW-Swim (−6–8%, *d+* 0.88–1.17, and −9–10%, *d+* 0.70–0.85, respectively) in comparison with EW-Sed. In addition, analysis of individual data revealed that all participants in EU-Swim and EW-Swim (four and seven, respectively) that exhibited elevated SBP at baseline returned to normal values, with no changes in the SBP status in EW-Sed (eight individuals).

As for blood biochemical analyses, in general, insulin and triglyceride levels were higher and HDL levels were lower among physically inactive controls than in the other groups. Above-normal levels of insulin and triglycerides were particularly observed in EW-Sed at baseline, which remained during the whole experiment. On average, the swimming intervention did not produce clear variations in those outcomes. However, it is worth noticing that in EW-Swim, six of eight children with elevated insulin levels at baseline presented normal values at the end of the intervention. In EW-Sed, this happened in only one out of eight participants, while the other four children with initial normal values showed hyperinsulinemia after 6 months. Finally, at baseline, 5 individuals in EW-Swim and 10 individuals in EW-Sed exhibited *acanthosis nigricans*. After 12 months, no change occurred in EW-Sed, while this condition disappeared in two children in EW-Swim.

At baseline, EU-Swim exhibited higher adiponectin and lower leptin compared to the other groups. No sufficient measurements could be taken for establishing a profile of TNF-α at baseline in EU-SWIM, but comparisons between 6 and 12 months in EW-Swim indicated that changes due to the swimming practice did not occur. The adiponectin concentration also remained mostly unaltered in all groups. On the other hand, EW-Swim showed a significant reduction in leptin levels (−14–18%, *d+* 0.29–0.41), while changes were not detected in the other groups.

The endothelial function was equivalent across groups at baseline, with values mostly falling within normal ranges. Improvements in % Increment were detected in EW-Swim (26–41%, *d+* 0.53–0.64). The LnRHI remained stable, except for the increase detected in EU-Swim (65–83%, *d+* 1.02–1.03). However, this group showed exceptionally reduced LnRHI at baseline.

Data of vagal modulation reflected by RR and RMSSD were mixed, and improvements due to swimming practice were unclear. The intra-abdominal and subcutaneous fat assessed through ultrasound was greater among children with excess vs. normal weight. In response to the intervention, SAT (−6%, *d+* 0.18) and VAT (−16%, *d+* 0.63) lowered in EW-Swim, while the other groups remained stable. All groups exhibited similar cIMT at baseline, and significant improvements were found only in EU-Swim (−25–28%, *d+* 1.66–2.15).

Table 3 presents data from the subgroup analysis including children with obesity extracted from EW-Swim and EW-Sed (e.g., OB-Swim and OB-Sed). Improvements in z-BMI (−22%, *d+* 1.43), W/H (−7–9%, *d+* 0.84–0.98), and cardiorespiratory performance (28–43%, *d+* 1.03–1.55) occurred in children with obesity that underwent the swimming intervention vs. physically inactive controls, despite the similar time spent on screen activities and physical activity level. The overall cardiometabolic risk also improved in OB-Swim vs. OB-Sed, as reflected by favorable adaptations in SBP (−7–12%, *d+* 1.25–2.03), DBP (−10–11%, *d+* 0.86–1.03), TNF-α (−17%, *d+* 1.15), adiponectin (22%, *d+* 0.40), leptin (−13–32%, *d+* 0.34–0.87), and abdominal fat distribution reflected by SAT (−13%, *d+* 0.42) and VAT (−21%, *d+* 0.94). Differences between groups in markers of endothelial function, autonomic modulation, and cIMT were not detected.

## 4. Discussion

This controlled trial originally investigated the effects of a 12-month recreational swimming intervention applied during the whole school year (6 + 3 months interspersed with a 3-month school break interval) on physical activity, cardiorespiratory fitness, and several cardiometabolic risk factors in school-age children and adolescents with normal and excess weight. Our data demonstrated that swimming twice a week was capable to induce favorable adaptations in health markers after 6 and 12 months of intervention, improving the fitness and overall cardiometabolic risk in participants with overweight or obese.

Increases in physical activity and fitness are associated with health benefits, including a reduction in cardiometabolic risk [30,31]. In pediatric age groups, prior studies reported improvements in cardiorespiratory capacity and some cardiometabolic risk markers in response to aerobic and resistance training, sports, or recreational activities, provided adequate intensity and volume are applied [10,11,32]. Those adaptations seem to occur regardless of changes in body mass or composition [33]. A previous controlled trial including obese children aged 11 ± 2 years showed improvements in body mass (−4 Kg), fat percentage (−5%), and cardiorespiratory fitness assessed using 20 m shuttle-run tests until fatigue (4 rounds or 10%) in swimming vs. control group [15]. In the present study, a reduction in z-BMI and W/H occurred in EW-Swim, which concurs with those data, but the magnitude of those changes did not account for the substantial improvements in Yoyo test performance among individuals with excess weight that underwent the swimming intervention vs. eutrophic or physically inactive controls. After 12 months, the distances covered in the Yoyo test by children and adolescents with excess weight who participated in the swimming activity approached the performance of the normal weight group. This period seemed to be sufficient for the adequate learning of swimming techniques, which probably allowed increasing the intensity of sessions to achieve moderate-to-vigorous training intensities as recommended by the WHO [34]. The increase in physical activity level was also greater in EW-Swim vs. EU-Swim. Since individuals with excess weight were less active at baseline, this finding was probably a direct consequence of swimming practice. On the other hand, screen time did not change in all groups, remaining above the current WHO recommendations [34].

In short, our data suggested that the relatively small increase in the weekly physical activity associated with the swimming intervention was sufficient to improve aerobic fitness, regardless of changes in screen time or body mass. This information is relevant for practitioners, reinforcing the independent impact of exercise on the work capacity and the need for alternatives capable to motivate children with excess weight to participate in regular physical training, even with a frequency of a few days per week.

Indices of abdominal obesity are better markers of cardiovascular risk than BMI or W/H in isolation [35]. The evaluation of abdominal fat distribution by ultrasound provides accurate and non-invasive information on fat location, identifying individuals at high cardiometabolic risk [28]. Evidence indicates that different modalities of regular exercise can improve body fat distribution in pediatric age groups [36]. Our findings show that after 12 months of swimming practice, participants with excess weight significantly reduced subcutaneous and intra-abdominal fat vs. physically inactive controls, thereby approaching the values of the eutrophic group. The decrease was 6–10% among individuals with excess weight and 13–21% among those classified as obese. We could not find previous studies comparing the effects of swimming on visceral adipose tissue in children with overweight or obese. On the other hand, a recent meta-analysis [37] suggested that regular exercise alone would be capable to reduce visceral adipose tissue, with better responses occurring as a function of regular exercise vs. caloric restriction alone. The optimal exercise strategy for lowering visceral adiposity in children and adolescents with excess weight is not clear and should be addressed in further research.

The hypotensive effects of exercise training seem to depend on the blood pressure at baseline [38]. Most of the individuals in our sample were normotensive, and less than 40% of participants had elevated blood pressure at baseline. Nonetheless, our data revealed a consistent reduction of 8–9 mmHg in SBP and DBP among swimmers regardless of the nutritional status, corresponding to approximately 8–10% of the initial values. In the swimming groups, four participants with normal weight, two overweight, and five with obesity had elevated blood pressure at baseline and none after six months of intervention. Strong evidence indicates that a reduction of such magnitude may prevent the development of cardiovascular disease [39]. These data are consistent with a meta-analysis suggesting that sports may induce favorable tensional adaptations in children with obesity exhibiting elevated blood pressure [40]. Of particular interest, at least one prior study [41] reported favorable changes in SBP of obese children that underwent a 24-week intervention including recreational aquatic activities.

A positive impact of exercise training on biochemical blood markers has been suggested [42]. Despite baseline fasting glucose levels being within the normal range on average, after 6 months of swimming intervention, there was a significant reduction in the excess weight group compared to the physically inactive controls, which is consistent with previous trials applying different types of exercise interventions, especially endurance training [10,42,43]. Insulin levels in participants with excess weight were in general above reference standards and higher vs. eutrophic participants. Although exercise-related changes in insulin resistance in children have not been fully elucidated, some studies reported improvements after relatively short exercise interventions [44,45,46]. Our findings about insulin are mixed. The comparison of average values suggested that swimming was not effective to reduce its levels. However, the analysis of individual data revealed that approximately 75% of excess weight participants with hyperinsulinemia at baseline normalized their values after the intervention, while in the physically inactive group, 80% of participants remained with high insulin levels. Additionally, *acanthosis nigricans*, which is frequent in the presence of hyperinsulinemia [47,48] was no longer observed in two out of five children allocated to EW-Swim. These findings are promising, but additional research is warranted to investigate whether recreational swimming may provoke favorable adaptations in the carbohydrate metabolism of children with excess weight.

Data on the effects of swimming on the lipid profile were inconclusive, which disagrees with previous research suggesting favorable exercise-related impacts on those outcomes [10,44,45]. It is well accepted that changes in LDL, HDL, or triglycerides are difficult to achieve with exercise training alone, particularly when their levels are within the normal range [42]. It is feasible to speculate that the overall exercise volume was not sufficient to provoke adaptations in the lipid profile in children and adolescents that exhibited relatively normal values at baseline. This is probably the first study assessing the effects of swimming over cytokines in a pediatric sample. Some cytokines are acknowledged to reflect the metabolic health and inflammation state [49]. Insulin resistance, for instance, has been often associated with the leptin-to-adiponectin ratio [50]. The effects of chronic exercise on leptin and adiponectin levels in children and adolescents with obesity are uncertain. Some studies reported increases in adiponectin and reductions in leptin [44,45,46], while others failed to detect changes [51]. Our findings add to the current knowledge by suggesting that a 12-month recreational swimming routine was effective to improve those outcomes. Decreases in leptin were observed in EW-Swim and OB-Swim, while adiponectin levels increased in OB-Swim vs. OB-Sed. These data are important as the regulation of leptin and adiponectin levels in the pediatric age group reduces the risk of early development of obesity-related chronic diseases [50].

TNF-α is a cytokine and an adipokine. As an adipokine, it promotes insulin resistance and is associated with obesity-induced type 2 diabetes [52]. Few studies addressed the TNF-α responses to exercise in pediatric populations with excess weight [53]. A trend of reduction has been described [10], but in general, trials failed to detect significant changes due to exclusive exercise intervention [46]. We originally demonstrated that a recreational swimming intervention may reduce the TNF-α levels, at least among children and adolescents with obesity (please refer to Table 3). Additional research is warranted to ratify these findings and to determine whether this would be an independent exercise-related effect or a consequence of changes in body mass or composition.

Cardiac autonomic dysfunction at early ages is known to increase cardiovascular risk during adulthood [54]. A predominance of parasympathetic cardiac modulation at rest is expected when there is no autonomic dysfunction, and accumulated evidence indicates that the vagal tone tends to be higher in active vs. inactive individuals of an equivalent age [55]. For this reason, investigating the potential impact of exercise training on autonomic control is relevant for children and adolescents. Our data for HRV indices of vagal modulation were inconclusive. Although RR changes indicate a slight increase in total HRV in EW-Swim, this finding did not correspond to greater RMSSD compared to controls. Previous studies investigating autonomic responses to physical training in children with excess weight showed improvements in HRV indices after interventions such as judo training [11] or recreational soccer [10]. The heterogeneity regarding exposures, outcomes, and sample characteristics increases the risk of bias in most available studies [55]. Despite this, moderate-to-vigorous PA seems to be positively associated with RMSSD [55], and our data agree with this assumption. Further research is needed to clarify the role of PA and notably swimming practice on HRV indices among children and adolescents with excess weight.

Obesity and high blood pressure contribute to the deterioration of vascular reactivity [56] and carotid obstruction in children [57]. The effects of swimming on vascular function in children and adolescents have been neglected by current research. A single trial investigated the impact of a 12-week intervention on vascular compliance among obese elementary children and reported improvement only on the right leg [15]. In our study, swimming was capable to improve the vasodilation capacity in EW-Swim, as suggested by an increase in percentage Increment (greater blood flow per minute). This is clinically important since a healthy vascular endothelium releases antioxidant and anti-inflammatory substances, preventing the development of atherosclerosis at early ages [58]. On the other hand, increased LnRHI was found only in EU-Swim. This could suggest that a favorable impact on reactive hyperemia due to swimming would be more likely to occur in individuals with normal vs. excess weight. However, as previously stated, the LnRHI at baseline seemed to be extraordinarily low in EU-Swim. This could be due to the size of the probes, which were perhaps too big for normal-weight children. Caution due to potential bias is, therefore, necessary in the interpretation of those findings.

Finally, we originally assessed the carotid thickness and abdominal fat through ultrasound measurements. Despite significant improvements in VAT and SAT in EW-Swim and OB-Swim, the cIMT remained unaltered in both groups. A reduction in the thickness of the carotid layer was detected only in EU-Swim. In brief, our data partially concur with previous studies reporting improvements in subclinical markers of atherosclerosis after 6 months of regular exercise in young individuals with obesity [41,59]. The characteristics of exercise training must be considered when comparing the present data with prior research—the intensity of our swimming program may have been lower than in trials applying aerobic [41] or concurrent training [51,59].

This study has strengths and limitations. Comparing children with overweight or obese vs. physically inactive and eutrophic controls is a differential of our protocol in comparison with prior experiments. Furthermore, this is probably the first controlled trial investigating the effects of a prolonged swimming intervention on a wide spectrum of cardiovascular risk factors in school-age children. The relatively small sample is the major limitation. However, controlled trials applying supervised extra-curricular exercise during long periods to children and adolescents are difficult to manage and, for this reason, scarce. It is important to mention the difficulties that participants faced to enroll in the experiment—financial, time, or mobility issues limited the availability of adults to take the children to the swimming classes. Other problems referred to the lack of assessments after the school break (to determine the impact of detraining) and the impossibility of evaluating a non-exercise control group with normal weight. Unfortunately, participants were not assessed when the swimming classes resumed after the summer school break, but the fact that changes did not occur in most outcomes between 6 and 12 months suggests that most adaptations occurred in the first phase of the intervention. The additional 12 weeks after the school break seemed enough to maintain the previous gains, which is interesting information for practitioners. Additional research shall determine whether the continuity of swimming practice throughout several years is capable to provoke indefinitely cumulative gains in physical fitness and cardiometabolic risk or whether a ceiling effect occurs in selected outcomes.

## 5. Conclusions

In conclusion, a recreational swimming program offered as an extra activity during the whole school year (6 + 3 months interspersed with a 3-month summer break) was capable to improve the cardiorespiratory fitness and overall cardiovascular risk of children and adolescents with overweight or obese. After 6 months, participants exhibited favorable changes vs. normal weight and physically inactive controls in anthropometric markers of nutritional status, estimated cardiorespiratory capacity, blood pressure, insulin profile, cytokines, endothelial function, and abdominal fat distribution. Significant changes in vagal autonomic modulation and carotid thickness were not detected. Our results ratify the hypothesis that recreational swimming might be an effective alternative to exercise programs designed to counteract cardiovascular risk in pediatric populations with excess weight. Additional research is needed to confirm these promising findings, assessing other cardiovascular risk factors and components of physical fitness in larger samples.

## Figures and Tables

**Figure 1 ijerph-19-17093-f001:**
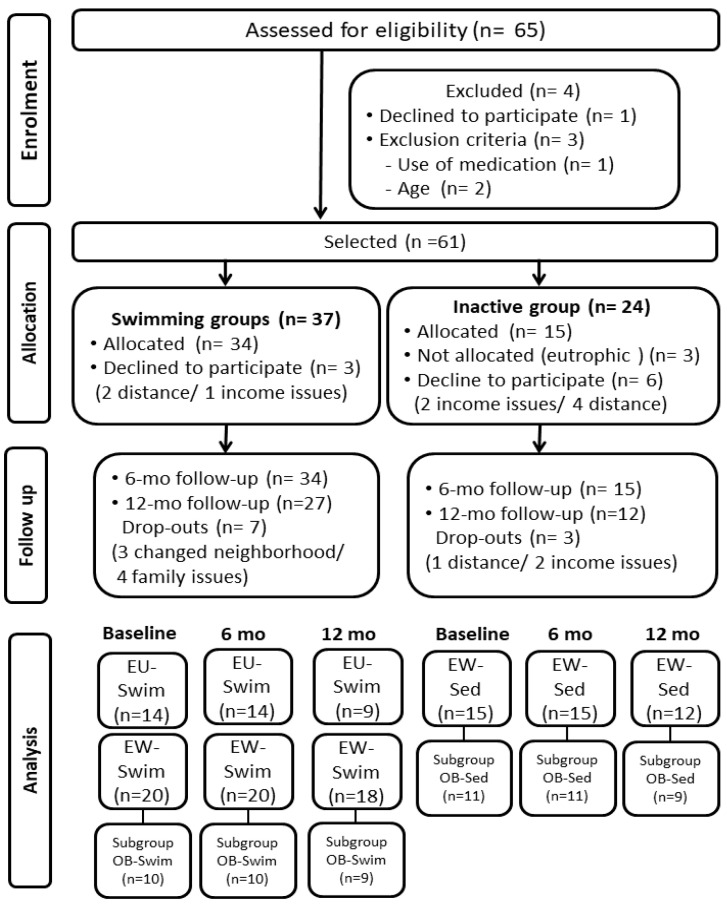
Flow diagram of the study.

**Figure 2 ijerph-19-17093-f002:**
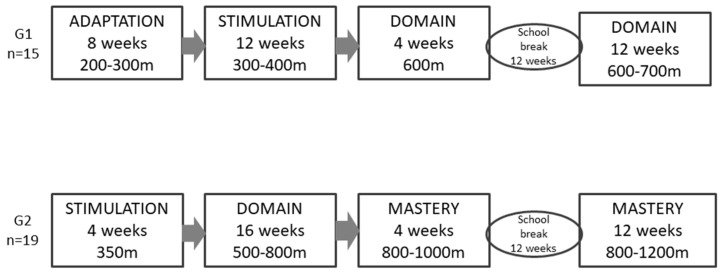
Groups of swimming development.

**Table 1 ijerph-19-17093-t001:** Sample characteristics at baseline in ‘eutrophic swimming’ (EU-Swim), ‘excess weight swimming’ (EW-Swim), and ‘excess weight sedentary’ (EW-Sed).

Outcome	EU-Swim	EW-Swim	EW-Sed
**N**	14	20	15
**Age (y)**	9.6 ± 1.5	10.3 ±1.8	10.9 ±2.1
**Female/male (n)**	9/5	11/9	10/5
**Nutritional status (n)**			
Eutrophic	14	0	0
Overweight	0	10	4
Obesity	0	10	11
**Pubertal stage (n)**			
Prepubertal (T1)	8	5	6
Beginning of puberty (T2 and T3)	4	12	5
End of puberty (T4 and T5)	2	3	4
** *Acanthose* ** ***nigricans* (yes/no)**	0/0	5/15	10/5
**Elevated blood pressure (≥90th percentile, n)**	4	7	8

Yes/No: number of individuals presenting or not *acanthoses nigricans.*

**Table 2 ijerph-19-17093-t002:** Data of anthropometry, physical activity level, screen time, cardiorespiratory fitness, blood pressure, biochemical markers, endothelial function, autonomic modulation, and ultrasound outcomes in ‘eutrophic swimming’ (EU-Swim), ‘excess weight swimming’ (EW-Swim), and ‘excess weight sedentary’ (EW-Sed) groups at baseline and after 6 and 12 months of intervention.

Outcome	EU-Swim			EW-Swim			EW-Sed		
	Baseline (n = 14)	6 Months (n = 14)	12 Months (n = 9)	Baseline (n = 20)	6 Months (n = 20)	12 Months (n = 18)	Baseline (n = 15)	6 Months (n = 15)	12 Months (n = 12)
z-BMI	0.26 ± 0.64 *^,ϕ,12^	0.26 ± 0.69 *^,ϕ^	0.10 ± 0.83	1.98 ± 0.69 ^ϕ, 12^	1.84 ± 0.64 *^,ϕ^	1.67 ± 0.48 ^ϕ^	2.91 ± 1.06	2.56 ± 0.56	2.95 ± 0.96
W/H	0.42 ± 0.02 *^,ϕ^	0.42 ± 0.02 *^,ϕ^	0.42 ± 0.03 ^ϕ^	0.51 ±0.06 ^ϕ, 6, 12^	0.48 ±0.04 ^ϕ^	0.47 ±0.04 ^ϕ^	0.57 ± 0.06	0.54 ± 0.03	0.58 ± 0.11
PAQ-C (score)	2.2 ± 0.5 ^6^	2.6 ± 0.5 *^,ϕ^	2.4 ± 0.6	1.9 ± 0.4 ^6, 12^	2.9 ± 0.5 ^ϕ, 12^	2.6 ± 0.4 ^ϕ^	2.1 ± 0.5	2.0 ± 0.5	2.0 ± 0.5
Screen time (h)	5.6 ± 2.3 ^6, 12^	4.6 ± 2.8	4.9 ± 3.5	3.4 ± 1.8	3.6 ± 1.9	3.9 ± 2.5	4.2 ± 1.9	4.0 ± 2.6	3.8 ± 1.8
Yoyo test (m)	506 ± 200 *^,ϕ, 6, 12^	641 ± 266 *^,ϕ, 12^	531 ± 93	337 ± 98 ^ϕ, 6, 12^	438 ± 117 ^ϕ^	474 ± 96 ^ϕ^	307 ± 108	338 ± 103	326 ± 117
SBP (mmHg)	106.4 ± 11.2 ^ϕ,6,12^	97.4 ± 7.8 ^ϕ^	97.1 ± 6.5 ^ϕ^	108.4 ± 8.5 ^ϕ,6,12^	100.3 ± 4.8 ^ϕ^	102.3 ± 4.8	118.2 ± 15.0 ^6,12^	110.2 ± 13.3	111.7 ± 14.4
DBP (mmHg)	60.6 ± 8.5 ^6^	54.4 ± 5.6	54.8 ± 5.2	62.2 ± 9.1 ^6,12^	56.9 ± 5.8 ^ϕ^	56.0 ± 4.9	64.4 ± 7.0	63.2 ± 13.4	63.9 ± 10.5
Glucose (mg/dL)	91.1 ± 6.8	91.4 ± 7.7 ^ϕ^	89.9 ± 7.5	94.1 ± 11.7	91.1 ± 5.3 ^ϕ^	89.8 ± 4.1	95.1 ± 7.7	97.1 ± 9.1	92.4 ± 7.7
Insulin (μlU/mL)	8.8 ± 4.2 ^ϕ^	9.2 ± 4.3 ^ϕ^	11.3 ± 5.1 ^ϕ^	17.5 ± 8.3 ^ϕ^	14.7 ± 7.6 ^ϕ^	17.9 ± 7.6 ^ϕ^	30.0 ± 8.7	29.9 ± 6.0	31.8 ± 3.2
TG (mg/dL)	93.2 ± 44.5 ^ϕ^	89.9 ± 59.0 ^ϕ^	85.6 ± 33.8 ^ϕ^	91.9 ± 35.3 ^ϕ^	93.1 ± 38.9 ^ϕ^	89.5 ± 26.7	153.9 ± 18.4 ^12^	174.6 ± 10.9	116.7 ± 64.8
HDL (mg/dL)	51.8 ± 10.7 ^ϕ^	48.5 ± 10.6 ^ϕ^	49.3 ± 7.7	50.3 ± 13.9 ^ϕ^	47.3 ± 8.5 ^ϕ^	47.9 ± 9.1	40.3 ± 6.9	35.2 ± 6.9	39.9 ± 6.0
LDL (mg/dL)	86.3 ± 17.3 ^12^	85.3 ± 25.5	99.0 ± 24.5	94.8 ± 26.0	83.1 ± 23.2 ^12^	100.6 ± 29.4	96.2 ± 20.4	82.2 ± 21.6 ^12^	105.2 ± 20.3
TNF-α (pg/mL)	-	-	-	1.07 ± 0.55	-	0.96 ± 0.21	1.06 ± 0.23	-	1.08 ± 0.12
Adiponectin (µg/mL)	10.3 ± 4.1 *^ϕ^	9.3 ± 3.2 ^12^	12.2 ± 5.1 ^ϕ^	7.9 ± 5.4	7.6 ± 4.8	8.3 ± 4.0	5.4 ± 3.5	5.5 ± 3.4	6.0 ± 3.7
Leptin (ng/mL)	11.0 ± 6.0 *^,ϕ^	11.6 ± 9.2 *^,ϕ^	14.1 ± 8.1 *^,ϕ^	26.4 ± 14.0 ^6,12^	22.8 ± 10.6 ^ϕ^	21.6 ± 9.1 ^ϕ^	37.7 ± 21.8	32.2 ± 22.9	42.3 ± 25.2
% Increment	323.9 ± 206.2 ^ϕ^	389.6 ± 123.9 ^12^	265.9 ± 112.3	238.2 ± 94.8 ^6,12^	334.6 ± 190.6	300.0 ± 136.8	209.3 ± 150.0	305.0 ± 156.3	347.1 ± 112.6
LnRHI	0.31 ± 0.14 *^,ϕ,6,12^	0.51 ± 0.24	0.57 ± 0.33	0.47 ± 0.24	0.58 ± 0.23	0.57 ± 0.22	0.54 ± 0.21	0.60 ± 0.17	0.64 ± 0.21
RR (ms)	818.6 ± 98.5	851.7 ± 89.8	826.0 ± 81.4	796.7 ± 68.9 ^6^	874.0 ± 121.9 ^12^	831.0 ± 133.6	847.4 ± 77.9	875.6 ± 77.9	910.6 ± 154.2
RMSSD (ms)	78.8 ± 30.7 ^ϕ^	82.6 ± 25.8 ^ϕ^	77.0 ± 20.1 ^ϕ^	68.7 ± 28.2 ^ϕ,6^	94.2 ± 43.2 ^12^	71.4 ± 25.6 ^ϕ^	92.4 ± 40.8 ^6^	110.9 ± 75.5	105.0 ± 57.2
SAT (cm)	1.32 ± 0.67 *^,ϕ^	-	2.88 ± 1.63	2.50 ± 1.03 ^12^	-	2.34 ± 0.71 ^ϕ^	3.35 ± 1.27	-	3.61 ± 1.13
VAT (cm)	2.71 ± 0.31 *^,ϕ^	-	3.09 ± 0.52 ^ϕ^	4.05 ± 1.08 ^12^	-	3.39 ± 1.01 ^ϕ^	4.61 ± 1.35	-	4.74 ± 0.91
cIMT right (cm)	0.060 ± 0.005 ^12^	-	0.043 ± 0.010 *^,ϕ^	0.057 ± 0.012	-	0.058 ± 0.011	0.061 ± 0.001	-	0.056 ± 0.010
cIMT left (cm)	0.063 ± 0.004 ^12^	-	0.047 ± 0.013 *^,ϕ^	0.058 ± 0.012 ^ϕ^	-	0.059 ± 0.011	0.064 ± 0.010	-	0.057 ± 0.011

z-BMI: BMI standard deviation score; W/H: weight-to-height ratio; PAQ-C: Physical Activity Questionnaire for Older Children; SBP: systolic blood pressure; DBP: diastolic blood pressure; TG: triglycerides; HDL: high-density lipoprotein; LDL: low-density lipoprotein; % Increment: percentage increase in flow per minute; LnRHI: natural logarithmic transformed reactive hyperemia-peripheral arterial tonometry index; RR: average of coupling intervals of all consecutive normal beats; RMSSD: square root of the mean of squares of the differences between successive normal RR intervals; SAT: thickness of subcutaneous fat; VAT: thickness of intra-abdominal fat; cIMT: thickness of the carotid middle intima layer. *: Significant difference *vs*. EW-Swim at a given month (*p* < 0.05). ϕ: Significant difference vs. EW-Sed at a given month (*p* < 0.05). Superscript numbers refer to within-group differences vs. indicated months (*p* < 0.05).

**Table 3 ijerph-19-17093-t003:** Data of anthropometry, physical activity level, screen time, cardiorespiratory fitness, blood pressure, biochemical markers, endothelial function, autonomic modulation, and ultrasound outcomes in ‘obese swimming’ (OB-Swim) and ‘obese sedentary’ (OB-Sed) groups at baseline and after 6 and 12 months of intervention.

Outcome	OB-Swim			OB-Sed		
	Baseline (n = 10)	6 Months (n=10)	12 Months (n = 9)	Baseline (n = 11)	6 Months (n = 11)	12 Months (n = 9)
z-BMI	2.56 ± 0.42 *^,12^	2.37 ± 0.39 *^,12^	2.00 ± 0.36*	3.29 ± 0.98	2.84 ± 0.53	3.24 ± 0.91
W/H	0.55 ± 0.06 ^6,12^	0.51 ± 0.03 *	0.50 ± 0.04 *	0.59 ± 0.05	0.57 ± 0.03	0.60 ± 0.11
PAQ-C (score)	1.9 ± 0.5 ^6^	2.8 ± 0.6 *^,12^	2.2 ± 0.4	2.1 ± 0.5	2.0 ± 0.5	2.1 ± 0.6
Screen time (h)	3.5 ± 2.0	3.7 ± 1.8	3.7 ± 2.9	4.4 ± 2.0	3.7 ± 2.6	4.0 ± 2.1
Yoyo test (m)	324 ± 101 ^6,12^	416 ± 75 ^12^	462 ± 75 *	290 ± 101 ^6,12^	331 ± 85	320 ± 113
SBP (mmHg)	111.8 ± 7.6 *^,6,12^	99.0 ± 4.6 *^,12^	104.1 ± 4.2 *	121.6 ± 15.0	118.0 ± 8.6	115.7 ± 12.5
DBP (mmHg)	62.8 ± 7.2 ^6,12^	56.5 ± 7.4 *	56.2 ± 5.5	66.2 ± 5.5	67.6 ± 15.0	65.9 ± 11.4
Glucose (mg/dL)	92.8 ± 5.1	91.1 ± 3.6 *	88.7 ± 4.9	92.7 ± 6.8	96.8 ± 10.2	90.6 ± 7.4
Insulin (μlU/mL)	19.9 ± 9.1	19.4 ± 9.0	19.5 ± 9.5	26.7 ± 11.4	24.3 ± 12.1	23.6 ± 11.9
TG (mg/dL)	107.8 ± 38.3	120.1 ± 38.1	93.2 ± 32.9	161.4 ± 119.3	212.3 ± 122.5	127.5 ± 71.8
HDL (mg/dL)	46.2 ± 12.5	46.3 ± 7.7	48.9 ± 7.3	39.0 ± 7.0	37.0 ± 7.7	40.3 ± 5.0
LDL (mg/dL)	103.8 ± 27.8	88.6 ± 23.8	111.1 ± 32.3	96.9 ± 23.4	89.2 ± 24.9	110.3 ± 17.3
TNF-α (pg/mL)	1.11 ± 0.15 ^12^	-	0.92 ± 0.18 *	1.07 ± 0.24	-	1.12 ± 0.09
Adiponectin (µg/mL)	5.7 ± 3.5 ^12^	5.8 ± 3.1 ^12^	7.0 ± 2.9 *	4.8 ± 3.1	5.0 ± 3.4	5.5 ± 3.2
Leptin (ng/mL)	33.8 ± 15.2 ^6,12^	29.3 ± 10.7 *	23.0 ± 8.9 *	43.8 ± 21.8	43.2 ± 18.6	45.0 ± 18.0
% Increment	207.8 ± 75.8	296.1 ± 119.2	282.8 ± 120.6	199.8 ± 143.4	292.4 ± 119.8	383.0 ± 100.5
LnRHI	0.46 ± 0.22	0.63 ± 0.21	0.52 ± 0.23	0.58 ± 0.21	0.62 ± 0.12	0.68 ± 0.19
RR (ms)	815.8 ± 79.5	930.3 ± 144.5	895.9 ± 144.9	855.4 ± 73.7	862.3 ± 137.8	945.5 ± 174.8
RMSSD (ms)	84.1 ± 28.5	114.9 ± 44.4	85.5 ± 22.4	89.6 ± 35.1	94.2 ± 65.7	95.7 ± 46.3
SAT (cm)	3.09 ± 1.10 ^12^	-	2.70 ± 0.70 *	3.54 ± 1.35	-	4.06 ± 1.06
VAT (cm)	4.65 ± 1.04 ^12^	-	3.69 ± 1.01	4.68 ± 1.27	-	4.44 ± 0.75
cIMT right (cm)	0.061 ± 0.008	-	0.059 ± 0.012	0.061 ± 0.003	-	0.057 ± 0.011
cIMT left (cm)	0.059 ± 0.009	-	0.061 ± 0.008	0.064 ± 0.006	-	0.059 ± 0.008

z-BMI: BMI standard deviation score; W/H: weight-to-height ratio; PAQ-C: Physical Activity Questionnaire for Older Children; SBP: systolic blood pressure; DBP: diastolic blood pressure; TG: triglycerides; HDL: high-density lipoprotein; LDL: low-density lipoprotein; % Increment: percentage increase in flow per minute; LnRHI: natural logarithmic transformed reactive hyperemia-peripheral arterial tonometry index; RR: average of coupling intervals of all consecutive normal beats; RMSSD: square root of the mean of squares of the differences between successive normal RR intervals; SAT: thickness of subcutaneous fat; VAT: thickness of intra-abdominal fat; cIMT: thickness of the carotid middle intima layer. *: Significant difference *vs*. OB-Sed at a given month (*p* < 0.05). Superscript numbers refer to within-group differences vs. indicated months (*p* < 0.05).

## Data Availability

Data are available upon reasonable request to the authors.

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
