# Peer review of "A Recreational Swimming Intervention during the Whole School Year Improves Fitness and Cardiometabolic Risk in Children and Adolescents with Overweight and Obesity"

_ijerph, 2022, doi:10.3390/ijerph192417093_

Round 1

Reviewer 1 Report

This paper is about monitoring physical activity, nutritional status, cardiorespiratory fitness, biochemical markers, and biological parameters in children for a year. Congratulations, to the authors for doing such laborious work. 

But from reading the paper some questions arise.

The first problem is that two tables exist with the same results (tables 2 and 3) and even some data such as TNF-a are missing to agree with the proposal in lines 287-8.

Also, the conclusion that the study is on low growth rate children (10.3±1.8 y), as seen at first glance in the abstract, does not represent the range of ages examined because adolescents and older biological children were included. The second observation is that while the intermediary assessment of physical activity shows an increase due to the extra swimming activity, this is not shown in the final measure which raises the concern that the children did not have this additional swimming activity.

In addition, here are a few more observations:

Ln: 106-8. It would be useful to report the distance per day covered at the beginning of the training period, at 6 months, at the restart, and at the end of the intervention. The improvement in the fitness and swimming technique of the children is certain to increase volume and speed in training over the same time period and we would expect this to be reflected towards the intermediary and end of the intervention.

Ln: 237. screen time: this definition appeared for the first time. You mentioned a few words in the methodology. Guedes' work is in Portuguese.  

Author Response

Rio de Janeiro, December 15th, 2022

-  Code: ijerph-2090110

- Title: One-year swimming intervention improves fitness and cardiometabolic risk in school-age children with overweight and obesity

Dear Editor,

Please find below our responses to the reviewers’ comments concerning our manuscript (ijerph-2090110) entitled “One-year swimming intervention improves fitness and cardiometabolic risk in school-age children with overweight and obesity”. Thank you for allowing us to re-submit our manuscript.

We have addressed all the issues raised by this reviewer. The manuscript has been rewritten according to the reviewer’s suggestions, and an itemized, point-by-point response to each of the comments has been provided. Changes in the manuscript are marked in yellow.

Yours Sincerely,

The Authors

*************************************************************************************

REVIEWER COMMENTS:

Comments and Suggestions for Authors

This paper is about monitoring physical activity, nutritional status, cardiorespiratory fitness, biochemical markers, and biological parameters in children for a year. Congratulations, to the authors for doing such laborious work.  But from reading the paper some questions arise.

1) The first problem is that two tables exist with the same results (tables 2 and 3) and even some data such as TNF-a are missing to agree with the proposal in lines 287-8.

Answer: Sorry for the oversight, Table 3 should depict data for the subgroup of obese participants. This was corrected and the text about TNF-alpha data is now consistent with the table. Thank you for this remark.

2) Also, the conclusion that the study is on low growth rate children (10.3±1.8 y), as seen at first glance in the abstract, does not represent the range of ages examined because adolescents and older biological children were included. The second observation is that while the intermediary assessment of physical activity shows an increase due to the extra swimming activity, this is not shown in the final measure which raises the concern that the children did not have this additional swimming activity.

Answer: We understand the reviewer’s concern. Albeit most of the participants were between 7 and 13 years old, the age range included adolescents until 16 years old. The abstract was amended to make clear that participants were school-age children and adolescents. It is worth mentioning that this precaution was already present in the conclusion of the manuscript, which reinforces the pertinence of the reviewer’s comment.

In addition, here are a few more observations:

3) Ln: 106-8. It would be useful to report the distance per day covered at the beginning of the training period, at 6 months, at the restart, and the end of the intervention. The improvement in the fitness and swimming technique of the children is certain to increase volume and speed in training over the same time period and we would expect this to be reflected towards the intermediary and end of the intervention.

Answer: We added the following information to the Methods (Subheading 2.2): “An initial assessment verified the level of aquatic development of participants to organize groups matched for their skills and ensure safety. The intensity of swimming classes was not directly measured, but they were designed to be of moderate- to vigorous intensity, in a progression that increased in direct proportion to the mastery of the techniques of the swimming styles, The covered distances in each class were recorded to provide insight on the exercise volume applied in each group and progression phase of the swimming program” (Lines 116-122). In addition, we included at the beginning of the Results the description of the stages performed by each initial swimming level group, including the duration and average distances covered in each one of them (Figure 2 and Lines 265-276).

4) Ln: 237. Screen time: this definition appeared for the first time. You mentioned a few words in the methodology. Guedes' work is in Portuguese.  

Answer: This is right and additional information was added to section 2.4 of Materials and Methods, as follows: “Screen time was also assessed by the PAQ-C, which includes a question about the average daily hours dedicated to any type of screen: TV, computer, video games, and especially smartphones. It is self-reported and participants responded with the help of their parents” (Lines 185-188).

In regards to Guede’s work, the ref [19] indeed presents the article in Portuguese by Guedes and Guedes (doi: 10.1590/1517-869220152106147594). We mentioned this reference because the Portuguese version was applied. This is now complemented by the original work to satisfy the reviewer’s demand: Crocker PRE, Bailey DA, Faulkner RA, Kowalski KC, Mcgrath R. Measuring general levels of physical activity: Preliminary evidence for the physical activity questionnaire for older children. Med Sci Sports Exerc. 1997;29(10):1344–9, doi: 10.1097/00005768-199710000-00011. Please, let us know if this is okay.

Reviewer 2 Report

Abstract, line 18. Why? Please include some rationale for the study.

Abstract, line 22. More detail about the swimming intervention would be useful.

Abstract, line 26. Who were the controls? It is not entirely clear from the descriptions of the groups above those that didn't do the intervention.

Introduction, line 69. Why 12 months?

Method, line 98. How were children recruited to the study? How was it advertised?

Method, lines 112-114. In this case, the study was not a a one-year intervention. This makes the title and some information in the last paragraph of the introduction misleading. Participants could have lost all or some of the adaptations gained in the first 6 months. Was there any consideration given to this? Ideally, participants should be been assessed again before the resumption of the training for the final 12 weeks in order to see if any adaptations had been maintained or lost. This is a major point that needs to be addressed.

Figure 1. I find this somewhat confusing with of participants in each group. It's not immediately clear how all the numbers in the analysis part add up and I had to read it several times before I could make sense of it. I would consider revising this.

Method, line 161. More info needed on blood sampling. Guessing venous, but not stated.

Method, lines 227-228. Please state the values used to interpret Cohen's d.

Results, line 237. First time screen time has been mentioned. Need to mention in methods.

Table 2. Why are there no TNF-alpha values for the EU-swim group? You mention this on line 287 but don't expand on why.

Tables 2 and 3 are the same.

I feel some figures illustrating some of the changes would be useful. Looking at the tables with lots of information in is quite daunting and not visually appealing.

Discussion, line 327. This is not true as the intervention itself was only 9 months.

Discussion, line 336-337. You haven't supplied information about the intensity of your own study.

Discussion, line 355. I don't see what screen time has to do with this. It is not a determinant of aerobic fitness.

Discussion, line 400. First mention of what acanthosis nigricans actually is. Could be defined earlier, at first mention.

Discussion, line 479-482. This needs expanding on. It is a major limitation and means that you cannot claim this is a one-year intervention. You need to change this throughout so that you are saying the intervention is 9 months.

Author Response

Rio de Janeiro, December 15th, 2022

-  Code: ijerph-2090110

- Title: One-year swimming intervention improves fitness and cardiometabolic risk in school-age children with overweight and obesity

Dear Editor,

Please find below our responses to the reviewers’ comments concerning our manuscript (ijerph-2090110) entitled “One-year swimming intervention improves fitness and cardiometabolic risk in school-age children with overweight and obesity”. Thank you for allowing us to re-submit our manuscript.

We have addressed all the issues raised by this reviewer. The manuscript has been rewritten according to the reviewer’s suggestions, and an itemized, point-by-point response to each of the comments has been provided. Changes in the manuscript are marked in yellow.

Yours Sincerely,

The Authors

*************************************************************************************

REVIEWER COMMENTS:

1) Abstract, line 18. Why? Please include some rationale for the study.

Answer: The major problem here is the limit of 200 words and the number of results to present in the Abstract. The background and objectives were amended as follows: “The benefits of swimming as a treatment for overweight children are undefined. We investigated the effects of recreational swimming on cardiometabolic risk in children/adolescents with normal and excess weight (Lines 18-20)”.

2) Abstract, line 22. More detail about the swimming intervention would be useful.

Answer: Additional detail was provided in the abstract (Lines 23-24), please let us know if this is sufficient, considering the limit of 200 words.

3) Abstract, line 26. Who were the controls? It is not entirely clear from the descriptions of the groups above those that didn't do the intervention.

Answer: Controls were physically inactive. This is now specified in the Abstract by indicating that this group was the EW-Sed (Line 27).

4) Introduction, line 69. Why 12 months?

Answer: The 12-month intervention was designed because previous studies applied substantially shorter interventions. Experiments with adults generally investigated interventions with aquatic exercises (not necessarily swimming) no longer than 12 weeks (please, refer to DOI: 10.26717/BJSTR.2021.33.005331). In the only experiment we could find that investigated the impact of aquatic exercise (not swimming) on the body composition, physical fitness, and vascular compliance in elementary school children (average 11 years old), the intervention lasted 12 weeks (please, refer to DOI: 10.12965/jer.140115). We aimed to verify whether prolonged swimming programs, typically performed by children and adolescents in actual life would be capable to induce additional effects on multiple markers related to physical fitness and cardiometabolic disease. We considered that this would increase the ecological validity of our data and that this original information would be useful for practitioners and medical doctors that usually indicate this sport modality for children and adolescents. For this reason, our protocol included a whole school year, during which the swimming intervention was offered as an extra activity. A brief comment in this sense was added in the Introduction section, just before the paragraph stating the objectives of the study (Lines 69-72).

5) Method, line 98. How were children recruited for the study? How was it advertised?

Answer: We added this information to the Methods (Subheading 2.1): “The participants were initially recruited at the pediatric outpatient clinic of a public hospital, and through leaflets advertising the swimming project distributed in the University facilities (Lines 93-95)”.

6) Method, lines 112-114. In this case, the study was not a one-year intervention. This makes the title and some information in the last paragraph of the introduction misleading. Participants could have lost all or some of the adaptations gained in the first 6 months. Was there any consideration given to this? Ideally, participants should be been assessed again before the resumption of the training for the final 12 weeks in order to see if any adaptations had been maintained or lost. This is a major point that needs to be addressed.

Answer: This is indeed a major limitation of our design, which was unavoidable due to the school break and changes in the participants’ daily routines. Our purpose was to offer swimming classes as an extra activity integrating the school year, and we made amendments to clarify this point throughout the manuscript. Unfortunately, participants were not assessed when resumed the swimming classes after the summer school break due to budget restrictions, but the fact that changes did not occur in most outcomes between 6 vs.12 months suggests that most adaptations occurred in the first phase of the intervention. The additional 12 weeks after the school break seemed to be enough to maintain the previous gains. This is interesting information for practitioners. Additional research shall determine whether the continuity of swimming practice throughout several years is capable to provoke indefinitely cumulative gains in physical fitness and cardiometabolic risk or whether a ceiling effect occurs in selected outcomes. This rationale was added at the end of the Discussion section (Lines 553-560), and the interruption due to the school summer break was highlighted as a limitation of our study. For the sake of transparency and clarity, we made amendments to the Title, Abstract, Introduction, and Conclusion. Please, let us know if this is sufficient to attend to your concern.

7) Figure 1. I find this somewhat confusing with of participants in each group. It's not immediately clear how all the numbers in the analysis part add up and I had to read it several times before I could make sense of it. I would consider revising this.

Answer: Figure 1 was amended to improve clarity on how the different subgroups were composed. Please, let us know if this is better now.

8) Method, line 161. More info needed on blood sampling. Guessing venous, but not stated.

Answer: The correction was made (Line 193).

9) Method, lines 227-228. Please state the values used to interpret Cohen's d.

Answer: The correction was made (Lines 260-261).

10) Results, line 237. First time screen time has been mentioned. Need to mention in methods.

Answer: The correction was made, please refer to the Subheading “2.4 Physical activity level and cardiorespiratory fitness” (Lines 185-188).

11) Table 2. Why are there no TNF-alpha values for the EU-swim group? You mention this on line 287 but don’t expand on why.

Answer: Unfortunately, kits were not available to measure TNF-alpha in participants of all groups and we chose to measure all participants with excess weight in swimming and inactive groups. That is why we stated that: “No sufficient measurements could be taken for establishing a profile of TNF-alpha at baseline in EU-Swim” (Lines 351-352). Please, let us know whether this is convenient and necessary to explicitly mention this information in the text.

12) Tables 2 and 3 are the same.

Answer: Sorry for the oversight, Table 3 should depict data for the subgroup of obese participants. This was corrected and the text about TNF-alpha data is now consistent with the table. Thank you for this remark.

13) I feel some figures illustrating some of the changes would be useful. Looking at the tables with lots of information is quite daunting and not visually appealing.

Answer: We understand the suggestion, but the major challenge here is to select which outcomes to show as figures. We have decided to present tables because the number of illustrations would be excessive, perhaps making it unfeasible to publish the study. However, we are not averse to the suggestion. Please, we would appreciate suggestions as to what results would you like to see presented in this way.

14) Discussion, line 327. This is not true as the intervention itself was only 9 months.

Answer: At the beginning of the Discussion, we highlighted this issue for the sake of transparency and clarity (Lines 398-399). This was also clarified in the Title, Abstract, Introduction, Methods, and Conclusion.

15) Discussion, lines 336-337. You haven't supplied information about the intensity of your own study.

Answer: The intensity of swimming classes was not directly measured, but they were designed to be of moderate- to vigorous intensity, in a progression that increased in direct proportion to the mastery of the techniques of the swimming styles. We added this information to the Methods (Subheading 2.2) and amended the referred sentence as follows: “This period seemed to be sufficient for the adequate learning of swimming techniques, which probably allowed increasing the intensity of sessions to achieve moderate- to vigorous training intensities as recommended by the WHO” (Lines 418-420). Please, let us know if you agree with this.

Moreover, we added the following information to the Methods (Subheading 2.2): “An initial assessment verified the level of aquatic development of participants to organize groups matched for their skills and ensure safety. The intensity of swimming classes was not directly measured, but they were designed to be of moderate- to vigorous intensity, in a progression that increased in direct proportion to the mastery of the techniques of the swimming styles, The covered distances in each class were recorded to provide insight on the exercise volume applied in each group and progression phase of the swimming program” (Lines 116-122). In addition, we included at the beginning of the Results the description of the stages performed by each initial swimming level group, including the duration and average distances covered in each one of them (Figure 2 and Lines 265-276).

16) Discussion, line 355. I don't see what screen time has to do with this. It is not a determinant of aerobic fitness.

Answer: We believe that a misunderstanding occurred here. Our comment regarding the screen time refers to the results of physical activity level, and not aerobic fitness. We respectfully ask the reviewer to reconsider this point.

17) Discussion, line 400. First mention of what acanthosis nigricans actually is. Could be defined earlier, at first mention.

Answer: The correction was made (Lines 177-178).

18) Discussion, lines 479-482. This needs expanding on. It is a major limitation and means that you cannot claim this is a one-year intervention. You need to change this throughout so that you are saying the intervention is 9 months.

Answer: The correction was made, as per the response to item 6. Again, thank you for this important remark. In addition, the Conclusion was amended as follows: “In conclusion, a recreational swimming program offered as an extra activity during the whole school year (6 + 3 months interspersed with a 3-month summer break) was capable to improve the cardiorespiratory fitness and overall cardiovascular risk of children and adolescents with overweight or obese” (Lines 562-565).

Round 2

Reviewer 1 Report

Dear authors, thank you for the appropriate responses. Your comments have answered my concerns. I wish you a merry Christmas and a happy new year.

Author Response

Comments and Suggestions for Authors

Dear authors, thank you for the appropriate responses. Your comments have answered my concerns. I wish you a merry Christmas and a happy new year.

Answer: Thank you for your precious comments that improved our manuscript. We also wish you a merry Christimas and splendid 2023.

Reviewer 2 Report

I am largely happy with the changes the authors have made and their responses to my comments. I have several minor things for the authors to address.

Figure 1 is still slightly confusing. If my understanding from your method section is correct, OB-swim is a subgroup of EW swim (same with OB-Sed and EW-Sed). This isn't mentioned anywhere in the figure but would contribute greatly to understanding it better.

Results, line 282-293. This belongs in the method section where you explain why these groups were created.

Table 1. Should it say "T2 e T3" or should it be "T2 and T3"? If so, make this change throughout the table.

Author Response

Rio de Janeiro, December 15th, 2022

-  Code: ijerph-2090110

- Title: One-year swimming intervention improves fitness and cardiometabolic risk in school-age children with overweight and obesity

Dear Editor,

Please find below our responses to the reviewers’ comments concerning our manuscript (ijerph-2090110) entitled “One-year swimming intervention improves fitness and cardiometabolic risk in school-age children with overweight and obesity”. Thank you for allowing us to re-submit our manuscript.

We have addressed all the issues raised by this reviewer. The manuscript has been rewritten according to the reviewer’s suggestions, and an itemized, point-by-point response to each of the comments has been provided. Changes in the manuscript are marked in yellow.

Yours Sincerely,

The Authors

Comments and Suggestions for Authors:

I am largely happy with the changes the authors have made and their responses to my comments. I have several minor things for the authors to address.

1) Figure 1 is still slightly confusing. If my understanding from your method section is correct, OB-swim is a subgroup of EW swim (same with OB-Sed and EW-Sed). This isn't mentioned anywhere in the figure but would contribute greatly to understanding it better.

Answer: The Figure was amended to make clear that OB-Swim and OB-Sed groups were subgroups of EW-Swim and EW-Sed -groups. Please, let us know whether this is better.

2) Results, line 282-293. This belongs in the method section where you explain why these groups were created.

Answer: The correction was made.

3) Table 1. Should it say "T2 e T3" or should it be "T2 and T3"? If so, make this change throughout the table.

Answer: The correction was made. Thank you for the several comments that undoubtedly improved our manuscript.